# Re-Entrant Conformation Transition in Hydrogels [note 1]

**DOI:** 10.3390/gels7030098

**Published:** 2021-07-20

**Authors:** Oguz Okay

**Affiliations:** Department of Chemistry, Istanbul Technical University, Maslak, 34469 Istanbul, Turkey; okayo@itu.edu.tr; Tel.: +90-212-285-3156

**Keywords:** re-entrant volume phase transition, cosolvency, stimuli responsive hydrogels, hydrophobically modified hydrogels, polymer–solvent interactions

## Abstract

Hydrogels are attractive materials not only for their tremendous applications but also for theoretical studies as they provide macroscopic monitoring of the conformation change of polymer chains. The pioneering theoretical work of Dusek predicting the discontinuous volume phase transition in gels followed by the experimental observation of Tanaka opened up a new area, called smart hydrogels, in the gel science. Many ionic hydrogels exhibit a discontinuous volume phase transition due to the change of the polymer–solvent interaction parameter *χ* depending on the external stimuli such as temperature, pH, composition of the solvent, etc. The observation of a discontinuous volume phase transition in nonionic hydrogels or organogels is still a challenging task as it requires a polymer–solvent system with a strong polymer concentration dependent *χ* parameter. Such an observation may open up the use of organogels as smart and hydrophobic soft materials. The re-entrant phenomenon first observed by Tanaka is another characteristic of stimuli responsive hydrogels in which they are frustrated between the swollen and collapsed states in a given solvent mixture. Thus, the hydrogel first collapses and then reswells if an environmental parameter is continuously increased. The re-entrant phenomenon of hydrogels in water–cosolvent mixtures is due to the competitive hydrogen-bonding and hydrophobic interactions leading to flow-in and flow-out of the cosolvent molecules through the hydrogel moving boundary as the composition of the solvent mixture is varied. The experimental results reviewed here show that a re-entrant conformation transition in hydrogels requires a hydrophobically modified hydrophilic network, and a moderate hydrogen-bonding cosolvent having competitive attractions with water and polymer. The re-entrant phenomenon may widen the applications of the hydrogels in mechanochemical transducers, switches, memories, and sensors.

## 1. Introduction

Hydrogels are physically or chemically cross-linked hydrophilic polymers swollen in water without dissolving. Because all the polymer chains in a hydrogel are interconnected to form a single big molecule on a macroscopic scale, the conformational changes in the network chains can be visualized by a change in the hydrogel volume [1]. Thus, a gel provides macroscopic monitoring of single polymer behavior via simple swelling measurements. The idea of the formation of a 3D polymer network starting from small molecules goes back to the early days of polymer science, when polystyrene started to appear as a commercial product. Although polystyrene is soluble in good solvents, its insolubility was sometimes observed at the beginning of 30’s [2]. This insolubility was explained by Staudinger and Heuer, due to the existence of an impurity in styrene, namely divinylbenzene, which became later a popular tetrafunctional cross-linker to produce organogels [3]. In 1935, they report that the polymerization of styrene in the presence of a small amount of divinylbenzene leads to the formation of a chemically cross-linked polymer network absorbing large quantities of good solvents [3]. Figure 1 shows 86-years-old images of a polystyrene network before and after swelling in benzene, and its structure proposed by Staudinger. This pioneering discovery opened up new avenues to the design of soft and smart polymeric gels capable of absorbing large quantities of solvent.

The hydrophilic analogues of Staudinger’s organogels called hydrogels were later developed and attracted significant interest due to their similarities to biological systems, and stimuli-responsivity in aqueous media. The theoretical fundamentals of the formation and properties of gels developed by Flory [4], and de Gennes [5], the first application of hydrogels as contact lenses by Wichterle [6], the prediction of the volume-phase transition in gels by Dusek [7], and the discovery of smart hydrogels by Tanaka [8] can be considered as significant milestones in the development of the hydrogel science. Tanaka was also the first to observe an unusual swelling feature of hydrogels called re-entrant phenomenon at which they first collapse but then reswell as an environmental parameter such as the solvent composition or temperature is monotonically varied [9].

This review mainly focusses on the re-entrant conformation transition in hydrogels. I have to mention that although the 1940s to 1980s marked the golden age for the significant discoveries in the gel science, they are generally neglected in the past decade. Therefore, I focus here on the main challenges in the field of swelling-deswelling and re-entrant transitions of stimuli responsive hydrogels.

## 2. Swelling of Hydrogels

A unique characteristic of biological and synthetic hydrogels is their swelling when exposed to aqueous solutions during which their volume increases to assume an equilibrium value. The swelling equilibrium is determined by at least two opposite forces acting on a hydrogel specimen [4]: those due to the entropically favorable mixing of polymer segments with solvent molecules (mixing force), and due to deformation of the network chains to a more elongated state (elastic force), which is entropically unfavorable. The elastic contribution depends on the network model used, i.e., the affine network, phantom network, or constrained junction models [10,11]. Experimental works show that the phantom model describes the behavior of swollen networks more appropriately. In case of ionic hydrogels, another force becomes also effective because of the nonuniform distribution of mobile counterions between the hydrogel and the solution (ionic force). The classical Flory–Rehner theory assumes that the osmotic pressure of a gel is the sum of these three contributions, and is equal to zero at swelling equilibrium [4]. For affine and phantom network models with tetrafunctional cross-links, the classical Flory–Rehner equation including the ideal Donnan equilibria can be given using Equation (1a), Equation (1b), respectively [4,10,11]:(1a)ln(1−υ2)+υ2+χ υ22+νeV1(υ213υ2o23−0.5υ2)−fi υ2=0
(1b)ln(1−υ2)+υ2+χ υ22+0.5 νeV1 υ213υ2o23−fi υ2=0
where $\upsilon_{2}^{0}$ and $\upsilon_{2}$ are the polymer volume fractions in the hydrogel after preparation and at swelling equilibrium, respectively, χ is the polymer–solvent interaction parameter, $\nu_{e}$ is the cross-link density, i.e., the number of elastically effective network chains per volume of dry network, $V_{1}$ is the volume of a polymer segment equals to that of the solvent, and *f_i_* is the effective charge density in the network. The first two terms in these equations represent the entropy and the third term the enthalpy of mixing of polymer and solvent. The fourth term represents the elastic response of the polymer network to the change in the gel volume, while the last term represents the osmotic pressure due to the charges in the gel network. It is important that the elastic term is not a function of only $\nu_{e}$ and $\upsilon_{2}$ as reported in many recent papers neglecting the history of gel formation. It is also a function of $\upsilon_{2}^{0}$ as long as the hydrogel is prepared in the presence of a good solvent or a miscible polymer. Equation (1a,b) qualitatively predict the swelling degree of the hydrogels from their cross-link density $\nu_{e}$ or vice versa, assuming that the *χ* parameter of the polymer–solvent system is known. It must be noted that the χ parameter in these equations plays the role of the environmental factors such as the temperature or pH, and hence its variation may lead to a volume phase transition in hydrogels [11], as will be discussed later. As predicted by Equation (1a,b), decreasing the cross-link density $\upsilon_{e}$, or increasing the charge density *f_i_* significantly decreases the polymer concentration $\upsilon_{2}$, i.e., increases the hydrogel volume *V*_eq_ (=$\;\upsilon_{2}^{o}/\upsilon_{2}$), which is the bases for the design of superabsorbent polymers (SAPs). These equations also predict that weak polyelectrolyte hydrogels with a pH-dependent charge density, or hydrophobically modified hydrogels with a temperature-dependent χ parameter undergo reversible volume changes in response to pH or temperature, respectively, and hence they constitute the group of stimuli responsive hydrogels. 

## 3. Swelling-Deswelling Transition of Stimuli Responsive Hydrogels

Stimuli responsive hydrogels exhibit a reversible change in their volume in response to an external stimulus such as pH, temperature, and ionic strength of the environment, light intensity, electric and magnetic field, chemical triggers, etc. This volume change may occur gradually (continuous), or drastically (discontinuous) as a first-order volume phase transition. The discontinuous volume phase transition in gels from swollen to collapsed states, similar to the coil-to-globule transition of a single polymer chain, was first predicted in 1968 by Dusek and Paterson [7,12]. They showed that, under certain conditions, the chemical potential of the solvent in a gel passes through two extremes when plotted against the polymer volume fraction $\upsilon_{2}\;$(Equation (1a,b)). These extremes correspond to three roots satisfying the equilibrium condition with the solvent (Figure 2a). This means that there are three phases in equilibrium, namely pure solvent, swollen hydrogel, and shrunken hydrogel. The shape of the curves in Figure 2a resembles the P-V isotherm of van der Waals gases below the critical temperature. Thus, the gels can also undergo a coil (swollen gel)-to-globule (collapsed gel) transition, which is analogous to the liquefaction of gases. Dusek and Paterson showed that a discontinuous phase transition in gels may occur when they are prepared in a diluted solution (low $\upsilon_{2}^{o}$) and have a high cross-link density $\nu_{e}$ (Figure 2a) [7]. Unfortunately, they have not included the effect of the charge density of the network chains in their calculations which would result in a discontinuous volume phase transition without affecting the cross-link density.

Indeed, 10 years later, Tanaka was the first to experimentally observe the predicted volume phase transition in the example of polyacrylamide (PAAm) hydrogels immersed in acetone-water mixtures, and thus opened up a new area in hydrogel science, called smart hydrogels [8]. Figure 2b was taken from the historical paper of Tanaka showing the equilibrium volume *V_eq_* of two PAAm hydrogel samples in acetone-water mixtures of various acetone contents. Filled and open symbols are the data for the samples cured in the gelation tubes for 30 (gel I) and 3 days (gel II), respectively, before the start of the swelling tests. The hydrogel cured for 3 days undergoes a continuous deswelling transition at around 39% acetone, while that cured for 30 days shows a discontinuous volume phase transition at 41% acetone. This first experimental observation of a discontinuous volume phase transition in hydrogels was explained by Tanaka, with increasing cross-link density of the hydrogel as the curing time is increased [8]. However, this explanation was incorrect. Tanaka [13], and Ilavsky [14] later observed that the cross-link density of PAAm hydrogels remains almost unchanged during curing while their charge density increases. For instance, Figure 2c shows the inverse volume *V_eq_*^−1^ (open symbols) and the modulus *G* (filled symbols) of PAAm hydrogels in acetone-water mixtures of various acetone contents [14]. The hydrogels were prepared as described by Tanaka using ammonium persulfate-N,N,N′,N′-tetramethylethylenediamine (TEMED) redox initiator system. As the curing time is increased from 0.13 to 97 days (A to H in the figures), the degree of ionization was found to increase from 0 to 5.2%. Simultaneously, the extent of the collapse and the acetone content in the solvent mixture at the volume transition increase. A discontinuous change in both swelling ratio and modulus appears at curing times longer than 6 days. Thus, the real reason was the ionization of PAAm hydrogels due to the presence of the basic TEMED accelerator at the gel preparation, partially converting acrylamide (AAm) units to acrylic acid (AAc) ones facilitating volume phase transition.

Tanaka’s work has stimulated a huge interest in the theoretical and experimental studies on the volume phase transition in hydrogels [1,15,16,17,18,19,20,21,22]. Because the additional ionic term in eq 1 significantly affects the swelling properties, the existence of ionic groups facilitates the volume phase transition in hydrogels. Many synthetic and biological hydrogels containing ionic groups exhibit a reversible discontinuous volume phase transition depending on the external stimuli such as temperature, pH, composition of the solvent, etc. [1,15,17,19,20,23,24,25]. Because the external stimuli affect the extent of polymer–solvent interactions, the volume phase transition occurs due to the change of the *χ* parameter of polymer–solvent system. From stimuli responsive hydrogels, the temperature sensitive ones have attracted considerable attention due both to fundamental and technological interests [15,26]. The existence of competitive hydrogen-bonding and hydrophobic interactions in hydrogels is mainly responsible for their temperature sensitivity. Thus, one may create temperature sensitive hydrogels by incorporating hydrophobic groups as side chains into a hydrophilic polymer network. For instance, although the degree of swelling of PAAm hydrogel is almost independent on the temperature, replacing one of the amide hydrogens of AAm repeat units with the hydrophobic N-isopropyl group results in the classical temperature-sensitive poly(N-isopropylacrylamide) (PNIPAM) hydrogel undergoing a volume phase transition at around its lower critical solution temperature (LCST), i.e., at around 34 °C. Below this temperature, the hydrogel is swollen and it deswells as the temperature is increased. Nonionic PNIPAM hydrogel exhibits a continuous volume change, whereas incorporation of charged groups into the gel network turns this transition to a discontinuous one [26]. Figure 3a shows one of the first phase diagrams of PNIPAM hydrogels containing ionic sodium acrylate (NaAAc) segments. The molar concentration of NaAAc varies between 0 to 128 mM at a fixed NIPAM concentration of 700 mM. Increasing charge density of the hydrogels drives both the transition temperature and the extent of the transition towards higher values [26].

The reason of the temperature sensitivity in hydrogels is the competition between hydrogen-bonding and hydrophobic interactions. At a low temperature, water molecules near the hydrophobic side groups are strongly hydrogen bonded so that the hydrogel swells in water. Increasing the temperature weakens the hydrogen bonds while strengthens the hydrophobic interactions between alkyl side groups in order to minimize the contact between the hydrophobic surface and water. Thus, heating a hydrogel induces a transition from swollen to collapsed state at a critical temperature. The volume phase transition temperature of the hydrogels can be varied by changing the hydrophobicity of the network chains. For instance, as the amount of the hydrophobic N-t-butylacrylamide (TBA) units in TBA/N,N-dimethylacrylamide (DMAA) copolymer is increased from 20 to 60 mol%, the LCST decreases from 80 to 20 °C [28]. LCST also decreases with increasing length of alkyl side chains of poly(N-alkylacrylamide) hydrogels [29].

## 4. Is a Discontinuous Volume Phase Transition Possible in Nonionic Hydrogels or Organogels?

One may ask whether a discontinuous volume phase transition can be observed in nonionic hydrogels or organogels? Although it was initially claimed that the nonionic PNIPAM hydrogels exhibit discontinuous volume phase transition by changing the temperature, or the solvent composition [30], it was later shown that this transition is close to the critical point but a continuous one [26]. To my knowledge, such a transition has not been observed before in nonionic hydrogels, as it requires an unrealistic high cross-link density limiting the swelling degree to a low level (Figure 2a). Thus, the existence of ionic segments on the polymer network seems to be essential for observing large differences between the swollen and collapsed states and hence, a discontinuous volume change of the hydrogel. However, it was later shown that the solvent–polymer interactions that strongly depend on the polymer concentration may also induce a discontinuous volume phase transition in nonionic gels [31,32]. The interaction parameter *χ* is given as a function of polymer volume fraction $\upsilon_{2}\;$ by,
(2)$$\chi =\chi_{1}+\chi_{2}\;\upsilon_{2}+\;\chi_{3}\;\upsilon_{2}{}^{2}\;+\ldots$$
where $\chi_{1}$, $\chi_{2}$, and $\chi_{3}$ are functions of the external parameters. For nonionic gels, the condition for a discontinuous volume phase transition in a solvent was predicted as $\chi_{1\;}\leq 1/2$ and $\chi_{2}>1/3$ [31]. Although this condition is rare for most of the polymer–solvent systems, poly(isobutylene) (PIB)-benzene system at 24.5 °C exhibits a strong concentration dependent *χ* parameter, given by [31,33],
(3)$$\chi =0.500+0.30\;\upsilon_{2}+\;0.3\;\upsilon_{2}{}^{2}\;$$

Another requirement for the discontinuous phase transition in gels is the existence of a large quantity of a good solvent during gelation to decrease the polymer concentration $\upsilon_{2}^{o}$ [7]. This means that the organogels should be prepared in a dilute homogeneous solution. In order to observe a volume phase transition in PIB organogels, they were prepared by solution cross-linking of butyl rubber, which is PIB containing 1.5 mol% isoprene units, in toluene using sulfur monochloride as a cross-linker at a low polymer volume fraction, similar to Tanaka, $\upsilon_{2}^{o}$ = 0.047 [27]. The organogels having four different cross-link densities corresponding to the molecular weight of the network chains $M_{c}$ = 380, 220, 79, and 58 kg·mol^−1^ were immersed in toluene/methanol mixtures of various compositions. Figure 3b shows the equilibrium volume of PIB gels in terms of the polymer volume fraction $\upsilon_{2}$ plotted against the toluene volume fraction [27]. Independent on the cross-link density, all organogels are in a swollen state in the solvent mixture with less than 3 vol% methanol while they rapidly shrink with increasing amount of methanol from 3 to 10 vol%. For example, the organogel with the lowest cross-link density ($M_{c}$ = 380 kg·mol^−1^, filled symbols) swells 61-fold in toluene while the addition of methanol as a poor cosolvent leads to a seven-fold abrupt decrease in the organogel volume (Figure 3b). This deswelling transition resembles that of ionic PAAm hydrogels in acetone/water mixtures. The shape of the swelling curve for the organogel with $M_{c}$ = 380 kg·mol^−1^ indicates that this gel is in a critical state. Indeed, solution of Equation (1b) for the PIB gel with $M_{c}$ = 380 kg·mol^−1^ for *χ* as a function of $\upsilon_{2}$ gives the relation [27],
(4)$$\chi =0.4995+0.3177\;\upsilon_{2}+0.3899\;\upsilon_{2}{}^{2}$$
indicating that the *χ* parameter in toluene/methanol mixture is strongly polymer concentration dependent, and close to the requirement $\chi_{1\;}\leq \;1/2$ and $\chi_{2}>1/3$ for a discontinuous volume phase transition. Thus, changing the toluene/methanol composition is analogous to changing the pH, temperature, etc. of the previous hydrogel experiments. Expanding the logarithmic term of Equation (1b) and substitution of Equation (2) gives [27],
(5)(χ1−1/2)υ22+(χ2−1/3)υ23+(χ3−1/4)υ24+0.5 νeV1 υ213 υ2o23=0

The first and second derivatives of Equation (5) with respect to $\upsilon_{2}$, together with Equation (5) are equal to zero at the critical point. Calculations show that this condition was satisfied at $\upsilon_{2}$ = 0.088 revealing that the organogel with $M_{c}$ = 380 kg·mol^−1^ passes very close to critical conditions around this point (indicated by the dashed red circle in Figure 3b).

In concluding this section, a discontinuous volume phase transition in nonionic hydrogels or organogels is still a challenging task, and requires experimental work on polymer–solvent systems exhibiting a strong concentration dependent polymer–solvent interaction parameter *χ*. The closeness to the critical point in PIB-toluene/methanol system suggests that such a transition can be observed using loosely cross-linked organogels prepared at a low polymer concentration by varying the *χ* parameter playing the role of the environmental variable [11]. Such an observation may open up the use of organogels as smart and hydrophobic soft materials.

## 5. Re-Entrant Swelling Behavior

Another characteristic of stimuli responsive hydrogels is the re-entrant phenomenon in which they are frustrated between the swollen and collapsed states in a given solvent mixture. Thus, the hydrogel first collapses, then reswells if the solvent composition is continuously varied. Re-entrant phenomenon was first observed by Tanaka and co-workers in 1984 in poly(N,N-diethylacrylamide (PDEAA) and PNIPAM hydrogels containing ionic NaAAc segments (Figure 4) [9]. The hydrogels in water/dimethylsulfoxide (DMSO) mixtures undergo a re-entrant volume phase transition between swollen and collapsed states when the DMSO content of the solvent mixture is monotonically varied. For instance, PDEAA hydrogel containing 14 wt % NaAAc undergoes discontinuous deswelling and reswelling transitions at 60 and 80 vol.% DMSO, respectively (filled circles in Figure 4a). PNIPAM hydrogel with 7 wt.% NaAAc exhibits a similar behavior except that the reswelling transition occurs continuously (open symbols in Figure 4b). Moreover, the extent of the upper volume phase transition (at high DMSO concentration) is smaller than the lower transition for both hydrogels, which was attributed to the lower polarizability of DMSO as compared to water decreasing the degree of ionization of the charged acrylate groups [9]. The re-entrant transition behavior can qualitatively be explained by Tanaka using eq 1a with a *χ* parameter that changes depending on the composition of the solvent mixture. By taking the charge density as an adjustable parameter, it was shown that the extent of volume changes during phase transition increases with increasing charge density while nonionic hydrogels undergo smooth swelling and deswelling transitions [9]. Several mechanisms have been proposed for the re-entrant behavior of hydrogels basing on the competitive attractive interactions between hydrophobically modified hydrophilic polymer, water, and cosolvent components, and selective enrichment of the cosolvent inside the hydrogel [34,35,36,37,38,39,40,41,42,43]. The re-entrant volume phase transition was observed in several hydrophobically modified hydrogels in water–organic solvent (cosolvent), or water–linear polymer mixtures. These include, (a) cationic PAAm hydrogel in acetone-water mixtures [44], (b) PNIPAM hydrogels in aqueous solutions of methanol [35,36], ethanol [35,36], propanol [36] and low molecular weight poly(ethylene glycol)s (PEGs) [45,46,47,48], (c) poly(N,N-dimethylacrylamide) (PDMAA) hydrogels in aqueous solutions of acetone, dioxane, THF, and t-butanol [49], (d) poly(methacrylic acid) hydrogels in aqueous solutions of PEG of molecular weight 6000 g/mol [50], and (e) TBA/AAm copolymer hydrogels based on N-t-butylacrylamide (TBA) and acrylamide (AAm) in aqueous solutions of DMSO, ethanol, and acetone [51,52].

The filled symbols in Figure 5a–c present examples of re-entrant volume phase transition in nonionic PNIPAM [46], nonionic PDMAA [49], and ionic TBA/AAm (60/40 by mole) hydrogels [52], respectively. Here, the equilibrium hydrogel volume (*V/V_w_*) normalized with respect to its volume in water is plotted against the solvent composition in terms of the volume fraction *ϕ* of the cosolvent, which is PEG-300, acetone, and DMSO for PNIPAM, PDMAA, and TBA/AAm, respectively. All the three hydrophobically modified hydrogels are in a swollen state at *ϕ* = 0 and 1, i.e., in pure water or in pure cosolvent, respectively, while they deswell when water and cosolvent are mixed. In contrast, nonionic PAAm hydrogels (open symbols in Figure 5) undergo a monotonically deswelling with rising cosolvent concentration *ϕ*. This highlights importance of the hydrophobic interactions between polymer–polymer and polymer–cosolvent components of the ternary system. Because such interactions are absent between PAAm and cosolvent, the deswelling of PAAm hydrogels occurs smoothly as *ϕ* is increased. Thus, a hydrophobic modification of the hydrogel network is a prerequisite for the re-entrant phenomenon.

Figure 5a shows that the volume *V*/*V*_w_ of nonionic PNIPAM hydrogel in aqueous PEG-300 solution first decreases and after attaining a minimum volume at an intermediate PEG content (*ϕ* = 0.6), it again increases [46]. A similar re-entrant phenomenon is observable for the nonionic PDMAA hydrogel in water–acetone mixtures except that the minimum in the gel volume shifts to a higher cosolvent content (*ϕ* = 0.95) [49]. In contrast to the nonionic hydrogels, TBA/AAm copolymer hydrogel containing 1 mol% ionic 2-acrylamido-2-methylpropane sulfonic acid (AMPS) units undergoes discontinuous deswelling and reswelling transitions in water–DMSO mixtures at *ϕ* = 0.32 and 0.75, respectively, between which it remains in a highly collapsed state (Figure 5c) [52]. Figure 6a,b present the equilibrium volume *V*_eq_ of TBA/AAm copolymer hydrogels containing various amounts of AMPS units in water–DMSO, and water–ethanol mixtures, respectively, plotted against *ϕ* [52]. The nonionic TBA/AAm hydrogel exhibits a smooth deswelling/reswelling transition while this occurs rapidly as a volume phase transition in ionic hydrogels. The volume change at the transition becomes larger and the collapsed plateau becomes narrower as the ionic group content of the hydrogel is increased. Moreover, replacing DMSO with ethanol as a cosolvent shifts the re-entrant transition to a lower cosolvent volume fraction *ϕ*, which is attributed to the stronger hydrogen-bonding between ethanol and water [52]. Miki et al. investigated the re-entrant transition behavior of nonionic PNIPAM hydrogels in aqueous solutions of methanol, ethanol, 1-propanol, and 2-propanol [36]. In methanol-water mixtures, the deswelling transition occurs at a methanol mole fraction *x*_c_ of 0.14 (*ϕ* = 0.27) while for all other water–cosolvent mixtures, this happens at *x*_c_ = 0.04 (*ϕ* = 0.12-0.15). Figure 6 also shows that both the ionic and nonionic TBA/AAm hydrogels attain similar volumes in their collapsed states. Thus, the ionic force due to the counterions in the ionic hydrogels is responsible for a large volume phase transition.

To understand the relative magnitude of the physical interactions during the re-entrant transition, the swelling behavior of a nonionic PDMAA hydrogel was investigated in eight different water–cosolvent mixtures [49]. Figure 7a presents the equilibrium volume *V_eq_* of the hydrogel plotted against the volume fractions *ϕ* of DMSO, dioxane, acetone, and THF in the outer aqueous solution. All the water–cosolvent mixtures, except the water–DMSO one, induce a strong re-entrant transition in PDMAA hydrogels. They start to deswell between *ϕ =* 0.4 and 0.65 until attaining a minimum volume at *ϕ* close to unity (0.97–0.99), followed by reswelling of the collapsed gels. Thus, when a small amount of water was added into acetone, dioxane, or THF, PDMAA hydrogel undergoes a collapse transition. These results also suggest that a trace amount of water in these organic solvents can be detected by monitoring the volume change of PDMAA hydrogels. Figure 7b compares the swelling behavior of a nonionic PDMAA hydrogel in aqueous solution of various alcohols [49]. In aqueous methanol or ethanol, the hydrogel is in the swollen state over the whole range of *ϕ*, while a re-entrant behavior appears in aqueous solutions of 1-propanol and t-butanol.

The different cosolvents used in Figure 7 to induce the re-entrant swelling behavior in nonionic PDMAA hydrogels have substantially different polar (δ_p_) and hydrogen-bonding (δ_h_) solubility parameters [49]. For instance, as the number of carbons in alcohols increases from one to four, the polar component δ_p_ decreases from 12.3 to 5.7 MPa^1/2^, revealing that the extent of re-entrant transition increases with decreasing the polarity of the alcohol as a cosolvent (Figure 7b). Moreover, the hydrogen-bonding (δ_h_) component of the solubility parameter of the cosolvents also affects significantly on the re-entrant behavior of PDMAA hydrogels. Figure 8a presents how the minimum hydrogel volume (V_min_) in water–cosolvent mixtures varies with the hydrogen-bonding component δ_h_ of the cosolvent. Note that a low V_min_ value means a more collapsed state and hence, a large re-entrant behavior for a given water–cosolvent mixture (Figure 7). Moderate hydrogen-bonding cosolvents, namely acetone, THF, and dioxane with δ_h_ = ~8 MPa^1/2^ exhibit a strong re-entrant transition as compared to the alcohols with δ_h_ = 15–23 MPa^1/2^. Another parameter affecting the re-entrant transition is the relative magnitude of cosolvent-water and cosolvent-PDMAA attractive interactions. This parameter is represented by the volume ratio V_solv_, which is the ratio of the hydrogel volume in pure cosolvent to that in pure water. Thus, V_solv_ = 1 means that water–PDMAA and cosolvent-PDMAA attractions are equal in magnitude. Figure 8b shows that V_solv_ = ~0.4 observed for acetone, THF, and dioxane generates the maximum extent of re-entrant transition in PDMAA hydrogels as compared to the alcohols with a V_solv_ value of around unity. Thus, a stronger attraction of the cosolvent by water as compared to PDMAA is required for the re-entrant transition. From the above findings, one may conclude that a strong re-entrant conformation transition in PDMAA hydrogel requires a moderate hydrogen-bonding cosolvent, and moderate hydrophobic interactions between cosolvent and PDMAA. This condition facilitates the movement of the cosolvent molecules through the PDMAA network leading to a large volume change in PDMAA gels. Note that, although PDMAA hydrogels is in the swollen state over the entire range of DMSO, methanol, and ethanol concentrations, PNIPAM and TBA/AAm hydrogels undergo re-entrant transitions in these solvent mixtures [35,36,51,52]. This is attributed to the higher hydrophobicity of PNIPAM and TBA/AAm networks in comparison to the PDMAA network producing stronger polymer–cosolvent and polymer–polymer hydrophobic interactions facilitating the re-entrant phenomenon.

The re-entrant transition phenomenon in hydrogels can be explained mechanistically by the competitive interactions between the hydrophobically modified hydrophilic polymer (shortly denoted as polymer), water, and cosolvent components of the ternary system, as schematically illustrated in Figure 9. The cosolvent, which is less polar than water, is attracted by both water and hydrophilic part of the polymer by hydrogen-bonding interactions. It is also attracted by the alkyl side chains of polymer via hydrophobic interactions. Moreover, alkyl side chains also attract each other to form hydrophobic associations in the hydrogels acting as physical cross-links. As a consequence, the concentration of cosolvent in the gel differs from that in the external solution. The ratio of the cosolvent concentration inside the gel to that in the solution is defined as the cosolvent partition parameter *φ* [49]. Thus, *φ* = 1 means equal cosolvent concentration inside and outside the hydrogel while *φ* = 0 means exclusion of cosolvent from the hydrogel.

At a low cosolvent concentration *ϕ*, water–polymer and water–cosolvent attractions in the solution through hydrogen–bonding interactions dominate over the cosolvent-polymer attractions so that the hydrogel is in swollen state but the concentration of the less polar cosolvent inside the gel is slightly lower than that in the solution, i.e., *φ* < 1 (Figure 9a). As *ϕ* is increased, the decrease in the polarity of the solvent mixture as well as the osmotic pressure arising from the concentration difference of oligomeric or polymeric cosolvents between inside and outside the hydrogel lead to the disruption of the hydrogen-bonds between polymer and water, and deswelling of the hydrogel, which is accompanied with an increase in *φ* (a → b in Figure 9). Simultaneously, intramolecular hydrophobic interactions between the alkyl side groups of polymer become stronger due to the increasing polymer concentration in the deswollen hydrogel volume promoting further shrinkage of the hydrogel. Thus, disruption of hydrogen-bonding interactions between water and polymer followed by intramolecular hydrophobic interactions within the hydrogel network leads to deswelling transition at an intermediate or high value of *ϕ* (b → c in Figure 9). In the shrunken state of the hydrogel, *φ* is larger than unity because of the PNIPAM–water–1-propanol system hydrophobic environment within the gel network attracting cosolvent molecules via hydrophobic interactions. Indeed, Ishidao et al. observed that the highest value of *φ* in PNIPAM–water–1-propanol system is reached just after the deswelling transition [53]. The existence of hydrophobic interactions between polymer and cosolvent can also be seen from the high swelling ratio of the hydrogels in pure cosolvent, i.e., at *ϕ* = 1 (Figure 5, Figure 6 and Figure 7). They all exhibit a much higher swelling ratio as compared to the PAAm reference, highlighting the effects of the side isopropyl, dimethyl, and t-butyl groups of PNIPAM, PDMAAm, and TBA/AAm, respectively. Moreover, increasing cosolvent content of the hydrogel with increasing *ϕ* leads to its reswelling due to the increasing gel volume providing the flow of both cosolvent and water from solution to the gel phase (c → d in Figure 9). This results in a decrease in *φ* to approach unity at *ϕ* = 1. As a consequence, the cosolvent component of the solvent mixture first flows from the solution to the gel phase but then from gel to solution phase as *ϕ* is monotonically increased. This nonmonotonic change of the cosolvent partition parameter *φ* with *ϕ* is characteristic for the re-entrant phenomenon. Moreover, deswelling behavior of ionic hydrogels in water–cosolvent mixtures is associated with the formation of ion pairs in the hydrogel network. The degree of ion pairing is closely related with the dielectric constant within the hydrogel; the lower the dielectric constant *ε*, the higher is the degree of ion pairing, i.e., the lower is the swelling degree of the hydrogel [54,55]. Thus, the cosolvents having a much lower *ε* as compared to water pronounce the hydrogel deswelling, and hence generate a strong re-entrant volume transition in ionic hydrogels as compared to nonionic ones [54].

The mechanistic picture described above was supported by calculations using the Flory–Rehner theory, focusing on how the cosolvent partition parameter *φ* varies with the composition of the solvent mixture [49]. In the following, the components of the hydrogel system are denoted by the subscript *i* where *i* = 1, 2, and 3 for water, polymer, and cosolvent solvent, respectively. The interaction parameters between these components are represented by $\chi_{ij}$ ($i\neq j,\;\chi_{ij}$ = $\chi_{ji}x_{i}/x_{j}$) where *x_i_* is the number of segments on the component *i*. It is assumed that the hydrogel is immersed in a solvent mixture of infinite volume, i.e., *ϕ* remains constant for a given solvent-cosolvent mixture. The equilibrium swelling of the hydrogel in water–cosolvent mixture is reached when the chemical potentials of each liquid component inside and outside the hydrogel are balanced. According to the Flory–Rehner theory, the following equations describe the thermodynamic equilibrium condition of a nonionic hydrogel in water–cosolvent mixture [49]:(6)$$ln\left(\frac{1-\upsilon_{2}-\upsilon_{3}}{1-\phi }\right)+\left(\upsilon_{2}+\upsilon_{3}-\phi \right)-\left(\upsilon_{3}-\phi \right)/y+\chi_{12}\upsilon_{2}{}^{2}+\chi_{13}\left(\upsilon_{3}{}^{2}-\phi^{2}\right)+\left(\chi_{12}+\chi_{13}-\chi_{23}\right)\upsilon_{2}\upsilon_{3}+N^{-1}\upsilon_{2}\left(\alpha^{2}-0.5\right)=0$$
(7)$$-ln\left(\frac{1-\upsilon_{2}-\upsilon_{3}}{1-\phi }\right)+\left(1/y\right)ln\left(\upsilon_{3}/\phi \right)-2\chi_{13}\left(\upsilon_{3}-\phi \right)-\left(\chi_{12}+\chi_{13}-\chi_{23}\right)\upsilon_{2}=0$$
where $\upsilon_{i}$ is the volume fraction of the component *i* in the hydrogel, *N* is the number of segments of the network chains ($N={\left(\upsilon_{e}\;V_{1}\right)}^{-1}$), *α* is the linear swelling ratio, i.e., $\alpha ={\left(\upsilon_{2}^{o}/\upsilon_{2}\right)}^{1/3}$, and *y = x*_3_*/x*_1_. The cosolvent partition parameter *φ* is given by [49],
(8)$$\varphi =\frac{\upsilon_{3}}{\phi \left(1-\upsilon_{2}\right)}$$

The details of calculations for nonionic PDMAA hydrogels in aqueous solutions of acetone, dioxane, and THF were given elsewhere [49]. Shortly, $\upsilon_{2}^{o}$ and *N* were determined experimentally as $\upsilon_{2}^{o}$ = 0.056 and *N* = 1.6 × 10^3^, respectively. The interaction parameters $\chi_{12}$ between the PDMAA network and water is $\chi_{12}=0.48+0.33\upsilon_{2}$ [56]. The interaction parameter $\chi_{23}$ between the PDMA network and cosolvent was estimated from the swelling ratios of PDMAA hydrogels in pure cosolvents, and found as 0.124, 0.107, and 0.116 for acetone, dioxane, and THF, respectively. The value *y* was calculated as 4.1, 4.7, and 4.5 for acetone, dioxane, and THF, respectively. The Equations (6)–(8) were then solved for *φ* and $\chi_{13}$ as a function of *ϕ* to reproduce $\upsilon_{2}$ and hence, the hydrogel volume V_eq_ (= $\upsilon_{2}^{o}/\upsilon_{2}$) in water–cosolvent mixtures, given in Figure 7a.

Calculation results are shown in Figure 10 where the water–cosolvent interaction parameter $\chi_{13}$ (red curves), and the partition parameter *φ* (green curves) for nonionic PDMAA hydrogels in aqueous solutions of acetone, THF, and dioxane are shown as a function of cosolvent volume fraction *ϕ* [49]. The blue symbols represent the experimentally determined hydrogel volumes V_eq_ in these solvent mixtures. All the solvent mixtures induce nonmonotonic change in the partition parameter *φ* with increasing *ϕ*, which is accompanied with the re-entrant volume phase transition. In contrast, water–cosolvent mixtures inducing no re-entrant transition exhibit a monotonic increase in *φ* over the whole range of *ϕ* [49]. At low values of *ϕ* (not shown), the hydrogel is in swollen state and the partition parameter *φ* monotonically increases with increasing *ϕ*, that is, the solution inside the hydrogel is enriched by the cosolvent. As *ϕ* is further increased, both the partition parameter *φ* and $\chi_{13}$ rapidly decrease with increasing *ϕ*, as indicated by the arrows in Figure 10. Thus, the cosolvent is enriched in the solution as *ϕ* is increased due to the predominant water–cosolvent attraction in the solvent mixture lowering both the hydrogel volume *V_eq_* and the partition parameter *φ*_._ Between *ϕ* = 0.85 and 0.95, $\chi_{13}$ rapidly decreases while the partition parameter *φ* increases, i.e., the solution inside the gel is enriched by the solvent as *ϕ* is increased. The opposite behavior of the partition parameter *φ* in this regime is due to the collapsed state of the hydrogel. High concentrations of both polymer segments and cosolvent inside and outside the hydrogel, respectively, increases the number of contacts between them so that cosolvent molecules reenter the collapsed gel phase. The reentrance of cosolvent molecules into the gel network, however, does not suffice to swell the gel due to the strong water–cosolvent attractions represented $\chi_{13}$, which still dominate the swelling process. For *ϕ* > 0.95, the collapsed gel starts to swell again due to the favorable PDMA–cosolvent interactions now dominating the swelling process. Thus, in this regime, the gain in energy due to the increased number of contacts between PDMA and cosolvent exceeds the attraction forces between water and cosolvent. As a result, $\chi_{13}$ increases on rising *ϕ* so that the volume of the network increases. At the same time, *φ* decreases again and approaches to unity due to the increase of the gel volume that facilitates the penetration of both water and cosolvent molecules into the gel network. Thus, the re-entrant volume transition is due to the nonmonotonic change of the cosolvent concentration in the hydrogel that happens due to the competing hydrogen-bonding and hydrophobic interactions.

## 6. Conclusions

Hydrogels are attractive materials not only for their tremendous applications but also for theoretical studies as they provide macroscopic monitoring of the conformation change of polymer chains. For instance, coil-to-globule transition of single polymer chains can be visualized by the discontinuous volume change of a stimuli responsive hydrogel. The hydrogels also teach us how the competitive physical interactions determine the material properties. The pioneering theoretical work of Dusek predicting the discontinuous volume phase transition in gels followed by the experimental observation of Tanaka opened up a new area, called smart hydrogels, in the gel science. Many ionic hydrogels exhibit a discontinuous volume phase transition due to the change of the polymer–solvent interaction parameter *χ* depending on the external stimuli such as temperature, pH, composition of the solvent, etc. The observation of a discontinuous volume phase transition in nonionic hydrogels or organogels is still a challenging task as it requires a polymer–solvent system with a strong polymer concentration dependent *χ* parameter. The studies on PIB-based organogel-toluene/methanol system suggests that this can be achieved at a low polymer concentration at the gel preparation, and at a low cross-link density by varying the *χ* parameter. Such an observation may open up the use of organogels as smart and hydrophobic soft materials [57]. Over the past three decades, smart materials based on volume phase transition hydrogels have found many applications as catalyst, smart membranes, actuators, sensors, drug delivery, and microfluidics [58]. For instance, point-of-care devices for glucose detection were developed based on the volume phase transition of hydrogels induced by glucose [59]. Because the response rate of the hydrogels is inversely proportional to their size, research directed toward volume phase transition also triggered development of several techniques for the preparation of small hydrogels, i.e., nano- and microgels [58,60], as well as cryogels with superfast responsivity [61]. Moreover, in addition to the thermodynamic models based on the Flory–Huggins theory, several models have been developed to describe the volume phase transition of hydrogels including the multiphasic mixture theory, transport models, and molecular dynamic simulations [60,62,63,64,65].

The re-entrant phenomenon first observed by Tanaka is another characteristic of stimuli responsive hydrogels in which they are frustrated between the swollen and collapsed states in a given solvent mixture. Thus, the hydrogel first collapses and then reswells if an environmental parameter is continuously increased. The re-entrant phenomenon of hydrogels in water–cosolvent mixtures is due to the competitive hydrogen-bonding and hydrophobic interactions leading to flow-in and flow-out of the cosolvent molecules through the hydrogel moving boundary as the composition of the solvent mixture is varied. The experimental results reviewed here show that a re-entrant conformation transition in hydrogels requires a moderate hydrogen-bonding cosolvent having competitive attractions with water and polymer. Moreover, a hydrophobic modification of the hydrogel network is a prerequisite for the re-entrant phenomenon to create competitive hydrogen-bonding and hydrophobic interactions, and hydrophobic associations in the hydrogels via alkyl side chains. The cosolvent concentration at the re-entrant transition and the extent of polymer collapse can be varied by tuning the extent of intermolecular interactions as well as by the charge density and the hydrophobic character of hydrophobically modified polymer network. The re-entrant phenomenon may widen the application areas of the hydrogels in mechanochemical transducers, switches, memories, and sensors. For instance, the addition of a small amount of water into acetone, dioxane, or THF induces a collapse transition in PDMAA gel, suggesting that the trace amount of water existing in these organic solvents can be detected by monitoring the change in the gel volume.

## Figures and Tables

**Figure 1 gels-07-00098-f001:**
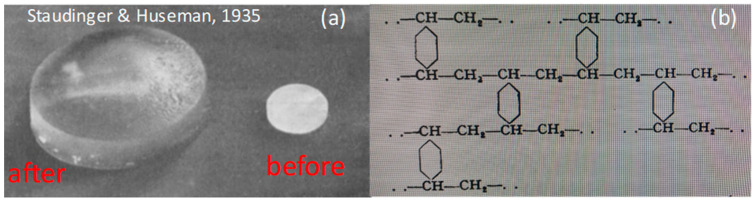
(**a**) Optical images of Staudinger’s organogels before and after swelling in benzene. (**b**) Chemical structure of cross-linked polystyrene proposed by Staudinger. From [3] with permission from Wiley-VCH Verlag GmbH & Co.

**Figure 2 gels-07-00098-f002:**
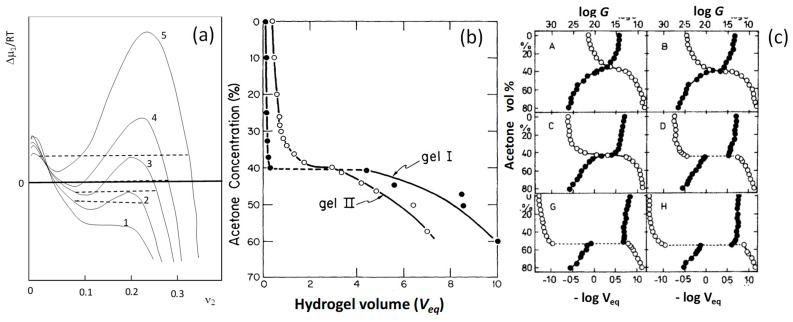
(**a**,**b**) The pioneering findings of Dusek (**a**) and Tanaka (**b**) on the volume phase transition in gels. (**a**) The dependence of the excess chemical potential ∆*μ*_1_ of the solvent on the polymer volume fraction $\upsilon_{2}$. The numbers 1 to 5 indicate increasing cross-link density. From [7] with permission from the John Wiley and Sons, Inc. (**b**) The volume *V_eq_* of PAAm hydrogels in acetone-water mixtures of various acetone contents. Filled and open symbols are the data for the gels cured in gelation tubes for 30 and 3 days, respectively. From [8] with permission from the American Physical Society. (**c**): Double-logarithmic plots showing the inverse of the hydrogel volume *V_eq_* (open symbols) and the modulus *G* in g·cm^−2^ (filled symbols) both plotted against the acetone concentration in acetone-water mixtures for PAAm hydrogels. The curing times are 0.13 (A), 3 (B), 6 (C), 12 (D), 78 (G), and 97 days (H). From [14] with permission from the American Chemical Society.

**Figure 3 gels-07-00098-f003:**
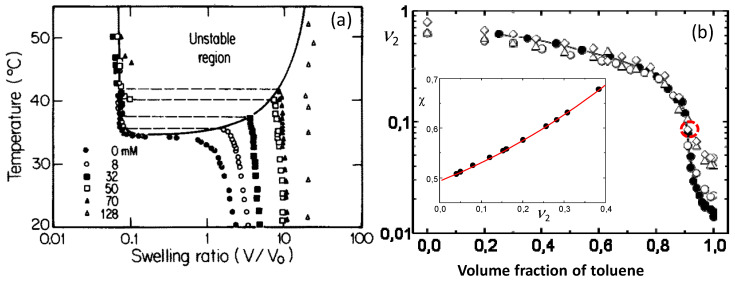
(**a**) Equilibrium swelling ratio (*V/V_o_*) of ionized PNIPAM hydrogels in water plotted against the temperature. The amount of ionizable sodium acrylate incorporated in 700 mM NIPAM is indicated. From [26] with permission from the American Institute of Physics. (**b**) The polymer volume fraction $\upsilon_{2}$ of PIB organogels in toluene/methanol mixtures shown as a function of the toluene content. PIB gels with the lowest crosslink density ($M_{c}$ = 380 kg·mol^−1^) are represented by the filled circles. The inset shows the dependence of χ parameter on the $\upsilon_{2}$ for PIB-toluene/methanol system. From [27] with permission from the American Chemical Society.

**Figure 4 gels-07-00098-f004:**
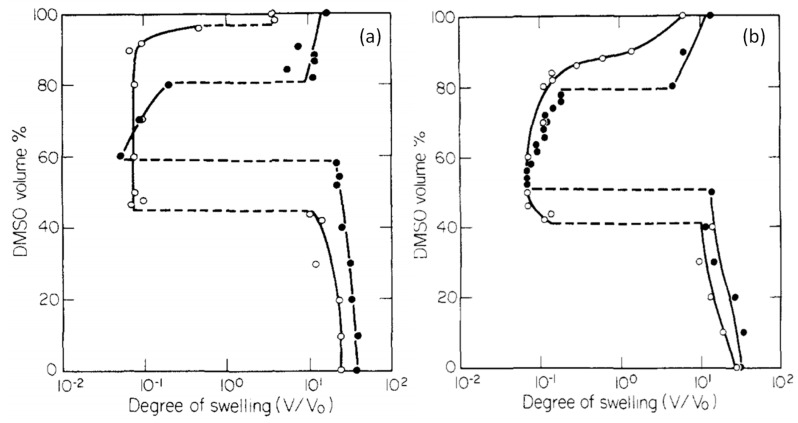
Equilibrium swelling degree (*V/V_o_*) of ionic PDEAA (**a**), and PNIPAM hydrogels (**b**) in DMSO/water mixtures of various compositions. The ionic comonomer (NaAAc) contents are 7 (open symbols) and 14 wt % (filled symbols). From [9] with permission from the American Chemical Society.

**Figure 5 gels-07-00098-f005:**
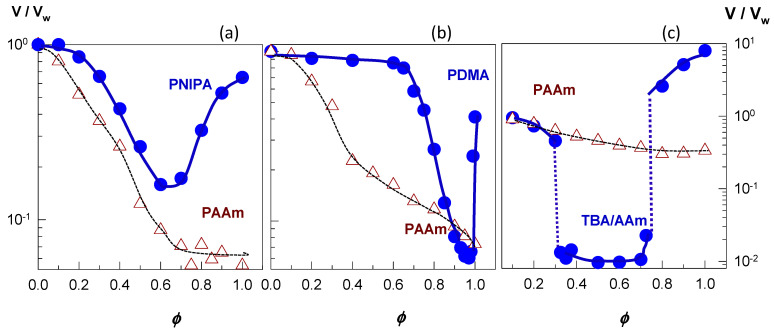
The swelling ratio V/V_w_ of three hydrophobically modified hydrogels in water–cosolvent mixtures plotted against the volume fraction *ϕ* of the cosolvent (filled symbols). For comparison, the swelling behavior of nonionic PAAm is shown by the open symbols. (**a**) Nonionic PNIPA hydrogel in water–PEG-300 mixture. Adapted from [46] with permission from the Wiley VCH Verlag GmbH, Weinheim. (**b**) Nonionic PDMAA hydrogel in water–acetone mixture. From [49] with permission from the Elsevier Ltd. (**c**) Ionic TBA/AAm (60/40 by mole) copolymer hydrogel in water–DMSO mixture. From [52] with permission from the Elsevier Ltd.

**Figure 6 gels-07-00098-f006:**
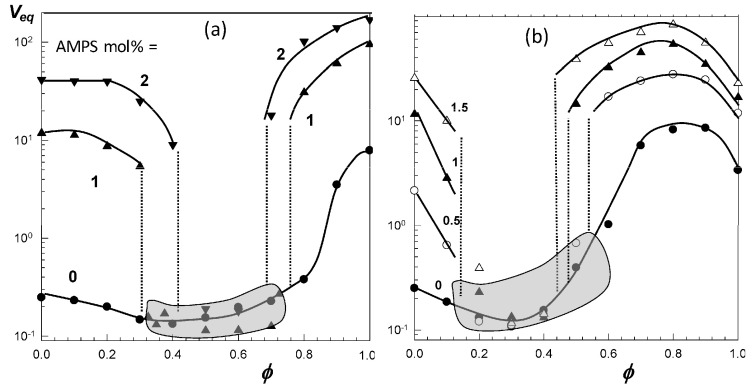
The volume V_eq_ of TBA/AAm (60/40 by mole) hydrogel in water–DMSO (**a**) and water–ethanol mixtures (**b**) plotted against the cosolvent volume fraction *ϕ* in the solution. AMPS contents (in mol% of the monomers) are indicated. The shaded areas present the collapsed plateau. From [52] with permission from the Elsevier Ltd.

**Figure 7 gels-07-00098-f007:**
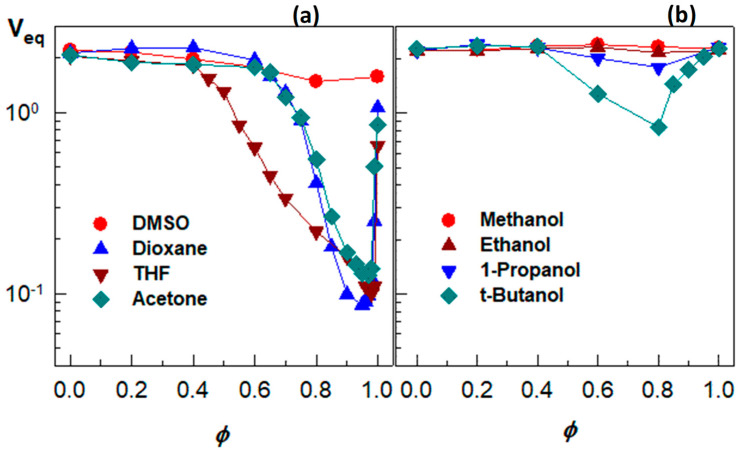
(**a**) the equilibrium volume V_eq_ of the hydrogel plotted against the volume fractions of *ϕ* DMSO, dioxane, acetone, and THF in the outer aqueous solution; (**b**) the swelling behavior of a nonionic PDMAA hydrogel in aqueous solution of various alcohol. From [49] with permission from the Elsevier Ltd.

**Figure 8 gels-07-00098-f008:**
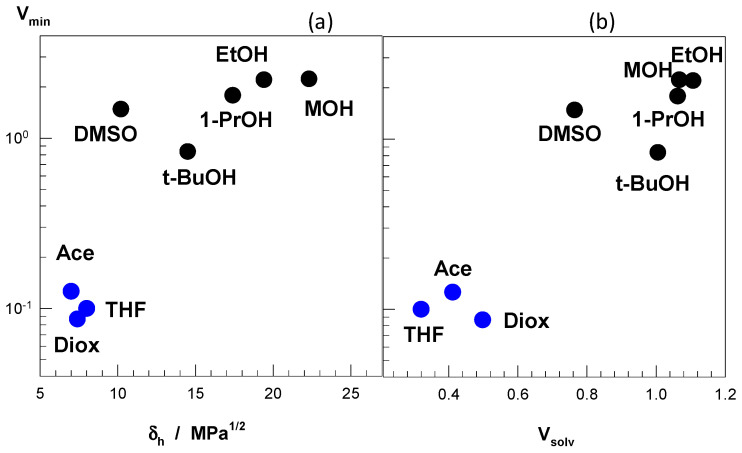
(**a**) The minimum volume (V_min_) of nonionic PDMAA hydrogels in water–cosolvent mixtures plotted against the hydrogen-bonding solubility parameter δ_h_ of the cosolvent. (**b**) The dependence of V_min_ on the swelling ratio V_solv_ of PDMAA hydrogels in the pure cosolvent normalized with respect to water. The cosolvents are indicated. MOH, methanol; EtOH, ethanol; 1-PrOH, 1-propanol; t-BuOH, t-butanol; Ace, acetone; Diox, 1,4-dioxane. From [49] with permission from the Elsevier Ltd.

**Figure 9 gels-07-00098-f009:**
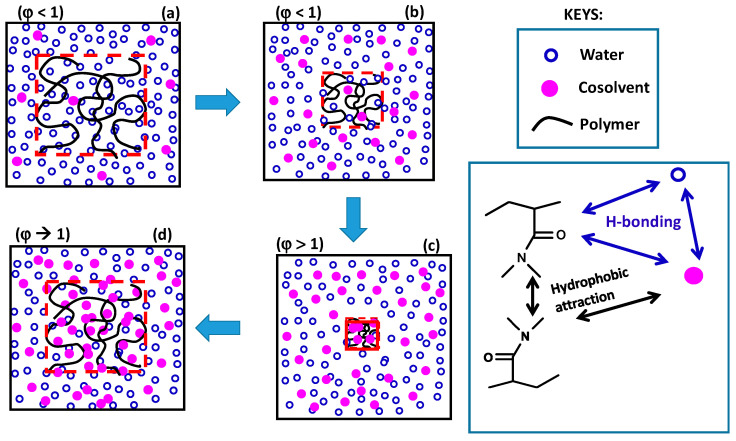
Schematic illustration of re-entrant volume phase transition in hydrogels (**a**–**d**). The double arrows indicate intermolecular interactions.

**Figure 10 gels-07-00098-f010:**
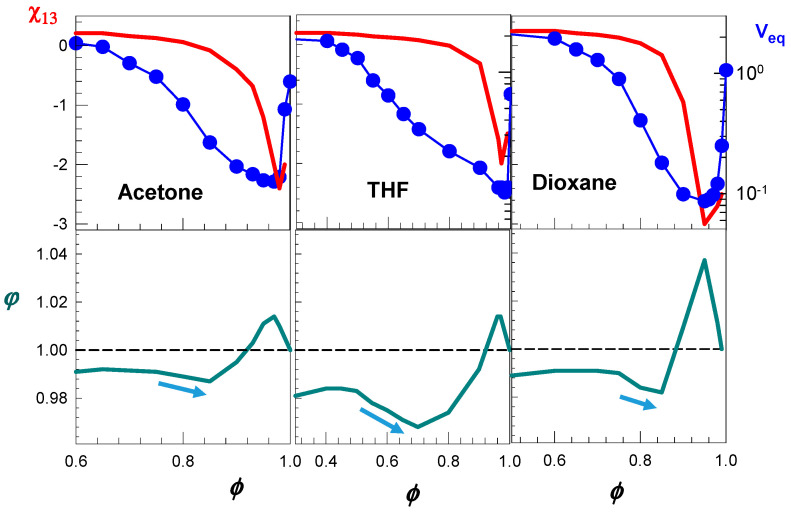
Interaction parameter $\chi_{13}$ (red curves), hydrogel volume *V*_eq_ (blue symbols), and the partition parameter *φ* (bottom curves) plotted against the cosolvent volume fraction *ϕ* in the water–cosolvent mixture. Calculations were using the Equations (6)–(8). The cosolvents are indicated. The horizontal dotted lines represent the condition *φ* = 1. Adapted from [49] with permission from the Elsevier Ltd.

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
