# Peer review of "Re-Entrant Conformation Transition in Hydrogelsâ€"

_gels, 2021, doi:10.3390/gels7030098_

Round 1
Reviewer 1 Report
The current review deals with the very interesting but not much reported reentrant properties of gels and hydrogels in cosolvent mixtures. This property is often misuindervealuated for some type of application and a review to remind the readership of Gels and thew scintific community working on organigel and hydrogel is surely more than welcome.
The mansucript is very well organized with a tiny part of history about the discovery of such phenomenon is highly appreciated. Such exploring work from Tanaka, a pioneer in the study of hydrogel, is more than welcome. The explanation of the phenomenon and the papers related to the topic that have shown the mechanisitic approach around such unexcpected behaviour are well explained.
The manuscript is actually warmly recommended for publication. although there is some minor issues regarding some table caption present in the manuscript that should be eliminated.
Author Response
I appreciate the time the reviewer took in reading our paper. I also thank the reviewer for his/her valuable remarks. I have tried to incorporate the reviewer’s comment into the revised paper as detailed below.
Comment: There is some minor issues regarding some table caption present in the manuscript that should be eliminated.
Response: We have to note that some important sentences, especially in the figure and table captions, in the submitted manuscript are missing in the file send to reviewers. They all were corrected as follows:
1) Page 2, 2nd paragraph
“Table 4. and de Gennes [5], …” replaced with:
“The hydrophilic analogues of Staudinger’s organogels called hydrogels were later developed and attracted significant interest due to their similarities to biological systems, and stimuli-responsivity in aqueous media. The theoretical fundamentals of the formation and properties of gels developed by Flory [4], and de Gennes [5],…”
2) Page 5, Figure 3 caption
“Table 700. mM NIPAM is indicated.” replaced with:
“The amount of ionizable sodium acrylate incorporated in 700 mM NIPAM is indicated”
3) Page 11, 2nd paragraph
“Table 9. The cosolvent, which is less polar than water, is attracted by…” replaced with:
“The reentrant transition phenomenon in hydrogels can be explained mechanistically by the competitive interactions between the hydrophobically modified hydrophilic polymer (shortly denoted as polymer), water, and cosolvent components of the ternary system, as schematically illustrated in Figure 9. The cosolvent, which is less polar than water, is attracted by …”
In addition, some minor corrections were made throughout the manuscript.
Best regards.
Oguz Okay
Reviewer 2 Report
This manuscript covers the theoretical background of hydrogel volume phase transition and reentrant conformation transition in hydrogels. These phenomena are critical in stimuli-responsive smart hydrogels that have various applications. The manuscript would considerable interest to those working in hydrogel design, biosensors, and drug delivery system design. In particular, the researchers working in smart sensors with controlled-release would be interested in the manuscript because the review describes the theoretical background of smart hydrogels. I would recommend that this paper needs minor revision to be published in Gels. I recommend the current manuscript should revise to include answers to the questions below:
- Please get the copyright of the figures before publication.
2. The manuscript focuses on the theoretical background of hydrogel volume phase transition and reentrant conformation transition in hydrogels. If the authors include recent research trends, it will help readers understand the meaning of the manuscript. Please introduce the state of the art of theoretical/experimental work of examples of hydrogel volume phase transition / reentrant conformation transition.
Author Response
I appreciate the time the reviewer took in reading our paper. I also thank the reviewer for his/her valuable remarks. I have tried to incorporate the reviewer’s comment into the revised paper as detailed below.
Comment: Please get the copyright of the figures before publication.
Response: I already have the copyright permissions for all figures and they were submitted together with the manuscript.
Comment: The manuscript focuses on the theoretical background of hydrogel volume phase transition and reentrant conformation transition in hydrogels. If the authors include recent research trends, it will help readers understand the meaning of the manuscript. Please introduce the state of the art of theoretical/experimental work of examples of hydrogel volume phase transition / reentrant conformation transition.
Response: The following paragraph and additional references were included as follows (page 15, lines 20-31, refs 57-65):
“Such an observation may open up the use of organogels as smart and hydrophobic soft materials [57]. Over the past three decades, smart materials based on volume phase transition hydrogels have found many applications as catalyst, smart membranes, actuators, sensors, drug delivery, and microfluidics [58]. For instance, point‑of‑care devices for glucose detection were developed based on the volume phase transition of hydrogels induced by glucose [59]. Because the response rate of the hydrogels is inversely proportional to their size, research directed toward volume phase transition also triggered development of several techniques for the preparation of small hydrogels, i.e., nano- and microgels [58,60], as well as cryogels with superfast responsivity [61]. Moreover, in addition to the thermodynamic models based on the Flory-Huggins theory, several models have been developed to describe the volume phase transition of hydrogels including the multiphasic mixture theory, transport models, and molecular dynamic simulations [60,62-65].”
In addition, some minor corrections were made throughout the manuscript.
Best regards.
Oguz Okay
Round 2
Reviewer 1 Report
The Author has addressed the corrections pointed out byt the 2 reviewers. The manuscritp is now recommended for publication in Gels.
Reviewer 2 Report
The author well revised the manuscript according to the review comments. I recommend the current manuscript for publication on Gels.